# Evaluation of the Feasibility of Harvest Optimisation of Soft-Shell Mud Crab (*Scylla paramamosain*) from the Perspective of Nutritional Values

**DOI:** 10.3390/foods12030583

**Published:** 2023-01-29

**Authors:** Weifeng Gao, Ye Yuan, Zhi Huang, Yongyi Chen, Wenxiao Cui, Yin Zhang, Hafiz Sohaib Ahmed Saqib, Shaopan Ye, Shengkang Li, Huaiping Zheng, Yueling Zhang, Mhd Ikhwanuddin, Hongyu Ma

**Affiliations:** 1Guangdong Provincial Key Laboratory of Marine Biotechnology, Shantou University, Shantou 515063, China; 2STU-UMT Joint Shellfish Research Laboratory, Shantou University, Shantou 515063, China; 3Institute of Tropical Aquaculture and Fisheries, Universiti Malaysia Terengganu, Kuala Terengganu 21030, Malaysia

**Keywords:** *Scylla paramamosain*, soft-shell crab, mineral composition, amino acids, fatty acids

## Abstract

Soft-shell crabs have attracted consumers’ attention due to their unique taste and nutritional value. To evaluate the feasibility of harvest optimisation of soft-shell mud crabs, the proximate composition, mineral composition, and total carotenoid, amino acid, and fatty acid contents of edible parts of male and female soft-shell mud crabs at different moulting stages were determined and compared from a nutritional value perspective. The results showed that the sex and moulting stages could significantly affect the nutritional values of the edible portions of soft-shell crabs. The female or male soft-shell crabs in the postmoult Ⅰ stage had a much richer mineral element content than that in other moulting stages. The total carotenoid content in female soft-shell crabs was significantly higher than that in male crabs in all moulting stages, while male soft-shell crabs had better performance in amino acid nutrition than female soft-shell crabs. Moreover, it was found that soft-shell crabs in the postmoult Ⅱ stage had significantly higher contents of eicosapentaenoic acid (EPA) and docosahexaenoic acid (DHA), while significantly lower contents of saturated fatty acids (SFA) than those in other stages. The present study will provide a reference basis for the diversified cultivation of soft-shell crabs, and further promote the development of the mud crab industry.

## 1. Introduction

Crab species such as the mud crab *Scylla paramamosain*, swimming crab *Portunus trituberculatus*, and snow crab *Chionoecetes opilio* are captured and cultivated all over the world as an important protein source [1,2,3]. In the traditional aquaculture or fishing industry, the crabs are usually sold or exported with their hard shells; however, over the past decades another form of “soft-shell crabs” has emerged as one of the most profitable novel forms for commercialisation in the crustacean farming industry. The term “soft-shell crabs” does not refer to a particular species of crab, but refers to the crab that has a newly moulted, decalcified, and soft exoskeleton [4]. It has been reported that many crab species can be cultivated for soft-shell crabs, including *Callinectes ornatus* [5], *Chionoecetes opilio* [6], *Portunus pelagicus* [7], *Portunus sanguinolentus* [8], *Scylla olivacea* [9], and *Scylla paramamosain* [10]. These soft-shell crabs have widely attracted consumers’ and researchers’ attention due to their high economic value, enriched nutrition, and unique mode of production [7,11].

Nutritional value, the core of aquatic products, is a comprehensive evaluation of protein, lipids, vitamins, minerals, amino acids, fatty acids, and other nutritional ingredients; its levels and compositions determine the degree of benefit to humans [3,12]. It greatly affects consumer preferences and purchasing behaviours, which ultimately determine the final commercial value of aquatic products; in other words, the public and the fisherman both prefer aquatic products with high nutritional value [13]. Various investigations reported that there were significant differences in the nutritional value of the edible portions of soft-shell and hard-shell crabs [4,8,12,14]. For example, a previous study on *Chionoecetes opilio* reported that soft-shell crabs had higher moisture and lower protein contents in muscle than hard-shell crabs [15]. In another study, Sudhakar et al. indicated that the protein, carbohydrate, and lipid contents in *Portunus sanguinolentus* were lower in soft-shell crabs as compared to hard-shell crabs [8]. However, the total contribution of minerals was lower in soft-shell crabs when compared to the hard-shell crabs [8]. Pathak et al. investigated the biochemical composition of soft- and hard-shell blue swimming crabs (*Portunus pelagicus*) and demonstrated that soft-shell crabs were rich in minerals, whereas hard-shell crabs contained more proteins, amino acids, and fatty acids [7]. Additionally, there were obvious differences in nutrient amounts between male and female crabs [16,17,18]. However, different moulting stages were also proved to be a great influential factor on the nutritional quality of the soft-shell crab [10].

Mud crabs *S. paramamosain* are being extensively cultured and have become one of the most important crustacean species farmed in the Asia–Pacific region due to their delicious taste, high nutritional value, and high market price [19,20]. The mud crab is a preferred candidate for soft-shell crab production in many countries around Asia due to its considerable economic value and better profit margins [7,9]. Additionally, the knowledge and culture technology of *S. paramamosain* for larvae, juveniles, and adults as well as their reproduction is mature and complete as compared to the other crabs, which facilitates the cultivation and production of soft-shell crabs from this species [21,22,23,24]. The current production of soft-shell crabs is based on the capture and storage of wild mud crabs (40–80 g) in the late postmoult, ecdysis, and premoult stages under controlled conditions until the onset of the next moulting with a body weight approximately (>80 g) [5,10,25]. If the soft-shell crabs are not collected soon after moulting, then they will lose their market as well as their nutritional value because the new exoskeleton hardens in a few hours [26]. So, to overcome such situations, it is important to collect the soft-shell crabs at the right time, before the shell hardens, and to incorporate advanced processing technologies and value addition to develop the appropriate products according to the actual needs of different consumers [9]. While culturing the soft-shell mud crabs, on-time collection is pertinent to ascertaining their market value and consumers’ choice; delayed collection strategies can lead to despair. Therefore, the purpose of this study was on the one hand to investigate the variation in nutritional value in male and female soft-shell mud crabs at different moulting stages, and on the other hand to identify the optimal harvesting time of the soft-shell mud crabs to further evaluate their potential as a functional food.

## 2. Materials and Methods

### 2.1. Ethics Statement

All experimental procedures in this study strictly complied with Chinese law pertaining to research on animals, and the protocol was performed with the approval of the Animal Ethics and Use Committee of Shantou University (Approval Code: 202301001, Approval Date: 8 January 2023).

### 2.2. Production Process of Soft-Shell Crab

Similar-sized and healthy mud crab juveniles (approximately 50 g) were obtained from a crab culture farm in Niutianyang (Shantou, Guangdong, China) located at N 23°22′23″, E 116°42′16″. After disinfecting for 10 min with a low concentration (50 ppm) of potassium permanganate solution (KMnO_4_), all crabs were transferred into 50 L plexiglass-walled aquariums with dimensions of 45 cm (length) × 32 (width) × 35 (height) in a recirculating aquaculture system (Zhongkehai Recirculating Aquaculture System, Qingdao, China) of the Marine Biology Institute (Shantou University, Shantou). Prior to the experiment, the crabs were acclimatised in the recirculation system for one week by feeding them fresh bait (razor clam) at the experimental conditions. After that, forty female crabs (initial weight: 52.97 ± 0.55 g; carapace width: 63.10 ± 0.46 mm; carapace length: 41.99 ± 0.40 mm) and forty male crabs (initial weight: 52.55 ± 0.39 g; carapace width: 64.08 ± 0.47 mm; carapace length: 41.88 ± 0.37 mm) were randomly distributed into eighty aquariums, respectively. During the experiment, the crabs were fed fresh razor clams once daily at 18:00 at a rate of 10% of their wet body weight. At 8:00 every morning, the uneaten bait and faeces were removed with a nylon screen. During the entire acclimation and experiment stages, seawater was continuously purified to maintain water quality through the physical and biofilter systems as well as the ultraviolet treatment, and the crabs were supplied with dissolved oxygen (DO) by air stones from air pumps. The seawater conditions were maintained as follows: water temperature at (28.5 ± 1.2) °C, salinity at (24.3 ± 1.4) g L^−1^, pH at 7.7 ± 0.4, total ammonia nitrogen (TAN) ≤ 0.5 mg L^−1^, and dissolved oxygen ≥ 6.0 mg L^−1^. The temperature was determined by an electronic thermometer, and other parameters were determined using a water quality testing kit (PanTian Bio-Tech, Xiamen, China). When the crab was found to be about to moult, it was immediately transferred to the dark box, then the moulting behaviour of the mud crab was observed and recorded through the underwater video camera (Barlus E3G2MPCX10, 1080 × 1920 pixels, Shenzhen Zhiyong Industrial Co., Ltd., Shenzhen, China) equipped with an LED (100 W, New Taipei, Taiwan, China) and a video capture subsystem (Intel^®^ Skylake-U I3-6100U, iVMS-4200H, 128GB storage memory, Hikvision, Hangzhou, China).

### 2.3. Experiment Design

In the present study, the moult stages of mud crabs were determined based on the previous study with slight modifications [27]. Here, we divided four moulting stages according to the experimental design as follows: premoult (hard-shell crab ready to moult), ecdysis (hard-shell crab just beginning to moult), postmoult Ⅰ (soft-shell crab 2 h after moulting), and postmoult Ⅱ (soft-shell crab 4 days after moulting). Whether female or male crabs, each moulting stage (premoult, ecdysis, postmoult Ⅰ, and postmoult Ⅱ) contained 10 crabs, respectively. The experiment aimed to collect samples at the end of each moulting stage and further analyse the nutritional values of female and male crabs at different moult periods

### 2.4. Sample Collection

Owing to the tissues (muscle and hepatopancreas) of the soft-shell crab being completely adhered together, the edible part of the soft-shell crab included almost the entire crab itself; hence, we used mixed tissues (muscle:hepatopancreas = 2:1) as the edible part of the soft-shell crab to facilitate subsequent detection and analysis. At the end of each moulting stage, a sample of approximately 3 g of fresh muscle from three crabs (1 g per crab) and 1.5 g of fresh hepatopancreas from three identical crabs (0.5 g per crab) were dissected, collected, and mixed as one sample (*n* = 3 per moulting stage) in a 5 mL microfuge tube, then stored immediately at −20 °C for further analysis. All sampling operations were performed on ice.

### 2.5. Determination of Proximate Composition

The proximate composition of the edible part of the soft-shell crab was determined and analysed following the procedures of the Association of Official Analytical Chemists [28]. Briefly, the moisture content was determined by drying the samples to a constant weight at 105 °C for 24 h using the oven-drying method. Crude protein content was calculated by determining nitrogen content (N × 6.25) using the Kjeldahl method with an automatic N analyser (Tecator Kjeltec autosampler system 1035 analyser, Foss Tecator, Sweden) after acid digestion in the Tecator Digestion System (Tecator digestor 2020, Foss Tecator, Sweden). Crude lipid content was determined by petroleum ether extraction using the Soxhlet method (Soxtec System HT6, Foss Tecator, Sweden). The ash content was determined using a muffle furnace at 550 °C for 24 h after carbonisation treatment. All determinations were performed in triplicate and the coefficient of variation was within 1.0%.

### 2.6. Determination of Mineral Composition

The mineral composition of the edible part of the soft-shell crab was determined and analysed by inductively coupled plasma–optical emission spectrometry (ICP–OES) (Thermo Fisher Scientific iCAP 7400 Duo; Waltham, MA, USA) following the method described by previous research with some modifications [29]. In brief, the edible part of the soft-shell crab was firstly homogenised and dried, then a 0.2 g sample (dry weight) was dissolved in a mixed solution of 10 mL of nitric acid (HNO_3_, 65%) in a 25 mL Teflon polytetrafluoroethylene (PTFE) tube, placed at room temperature for acidification overnight. After that, the solution was microwave-digested in a microwave digestion instrument (MARS 6, CEM, USA). The microwave digester temperature program was as follows: 190 °C with a 20 °C ramp from room temperature, held at 190 °C for 30 min, then followed by cooling until the temperature reached below 90 °C. The digested sample was diluted with 50 mL ultrapure water and stored at 4 °C for further analysis. The ICP-OES (iCAP 7400 Duo) was used to determine the concentration of minerals including Ca, Mg, Zn, Fe, Cu, Mn, and Se in the edible part of the soft-shell crab. All determinations were performed in triplicate and the coefficient of variation was within 1.0%.

### 2.7. Extraction and Determination of Total Carotenoids

The total carotenoids were extracted and determined by following the previous study with some modifications [29,30]. Briefly, approximately 300 mg freeze-dried homogenised samples (fine powder) were added to 8 mL of acetone and shaken at 200 rpm for 8 h in the dark at room temperature. Next, the samples were centrifuged at 5000 rpm for 5 min, then the absorption of the supernatant was measured at 480 nm in a UV-vis recording spectrophotometer (UV2501PC, Japan) to calculate the total carotenoid content with the extinction coefficient E (1%, 1 cm) of 1.900 [30]. All determinations were performed in triplicate and the coefficient of variation was within 1.0%.

### 2.8. Determination of Amino Acid Composition

The amino acid composition of the soft- and hard-shell crabs were determined and analysed using an S-433D automatic amino acid analyser (Sykam, Eresing, Germany) following the method described previously with slight modification [31]. Firstly, approximately 50 mg freeze-dried homogenised samples were successively weighed, placed into a 10 mL threaded screw-neck vial (CNW, Dusseldorf, Germany), and hydrolysed in 5 mL HCl solution (6 N) at 110 °C in a sand bath for 24 h under an N_2_ atmosphere. Next, the hydrolysate was cooled to room temperature and washed into a 50 mL volumetric flask with ultrapure water. Then, 1 mL of the solution above was transferred into a 4 mL ampoule bottle (CNW, Dusseldorf, Germany) and reduced to dryness using a termovap sample concentrator (MIULAB NDK200-1N, Hangzhou, Zhejiang, China). Afterward, the solvent was resuspended with 1 mL ultrapure water in an ampoule bottle and filtered through a 0.22 μm membrane (CNW, Dusseldorf, Germany). Finally, 20 μL of the solution were used for amino acid determination and the results were expressed as g 100 g^−1^ (dry matter) with all determinations performed in triplicate, and the coefficient of variation was within 1.0%.

### 2.9. Determination of Fatty Acid Composition

The fatty acid compositions of samples were determined according to the previous methods in the CLASS-GC10 GC workstation (Shimadzu, Kyoto, Japan) with minor modifications [32,33]. Briefly, approximately 500 mg of homogenised freeze-dried sample were added to a 12 mL glass screwed tube with 6 mL CH_3_Cl-CH_3_OH (2:1) solution containing 0.005% (*w*/*v*) 2,6-di-*tert*-butyl-4-methylphenol (BHT) as an antioxidant to prevent the oxidation of PUFA. After vortexing for 2 min, samples were left at 4 °C for 48 h. Next, the solution was centrifuged at 3000 rpm at 4 °C for 5 min and the supernatant was transferred to a new centrifuge tube. The solution in the tube was reduced to dryness using a termovap sample concentrator. Then, 2 mL KOH-CH_3_OH (0.5 N) solution were added to the tube, the sample was incubated in a water bath at 65 °C for 1 h and the tube was filled with N_2_. After cooling to room temperature, 2 mL BF_3_-CH_3_OH (14%) solution were added to the tube and incubated in a water bath at 70 °C for a further 30 min to produce the fatty acid methyl esters (FAMEs). Finally, 2 mL *n*-hexane and 0.5 mL saturated sodium chloride solution were added to the tube, shaken vigorously for 1 min, centrifuged at 3000 rpm at 4 °C for 5 min, and the supernatant was filtered through a 0.22 μm ultrafiltration membrane (CNW, Dusseldorf, Germany). The FAME solution in the ampoule was reduced to dryness by termovap sample concentrator, and the FAMEs were resuspended in 500 μL of *n*-hexane and stored at −20 °C until injected into a GC-17A gas chromatograph equipped with an autosampler and a flame-ionisation detector (Shimadzu, Kyoto, Japan). Individual fatty acids were identified by comparing them with known standards (Sigma Chemicals Co., St Louis, MO, USA, 99% purity). The quantification of individual fatty acids (% total fatty acids) was conducted using the CLASS-GC10 GC workstation (Shimadzu, Kyoto, Japan).

### 2.10. Statistical Analysis

The results were presented as means ± SE (n = 3). The differences among treatments were analysed by two-way ANOVA with sex (male and female) and moult stages (premoult, ecdysis, postmoult Ⅰ, and postmoult Ⅱ) as the first and second independent variables, respectively (SPSS 22.0, IBM Corp, New York, NY, USA). Differences were considered statistically significant at values of * *p* < 0.05, ** *p* < 0.01, and *** *p* < 0.001. GraphPad Prism 8.0 (San Diego, CA, USA) software was used for the processing of the histogram. The principal component analysis (PCA) of amino acids or fatty acids detected from the edible part was conducted using SIMCA-P+ software (Version 11.0.0.0, Umetrics AB, Malmo, Sweden).

## 3. Results and Discussion

### 3.1. Production Process of Soft-Shell Crab

The production process of soft-shell crabs is shown in Figure 1. Firstly, the activity of the mud crab was reduced before moulting (Figure 1I). When entering the moulting stage, there were cracks between the carapace and the abdominal carapace (Figure 1Ⅱ). Then, these cracks were further expanded with the crab’s moulting process. Meanwhile, the junction between the back end of the carapace and the abdomen was completely opened, and the mud crab moulted from its old exoskeleton through contractions and swings of the body (Figure 1Ⅲ–Ⅷ). Finally, the mud crab completely came out of the old exoskeleton and completed the process of moulting. The newly originated shell was soft, and based on this soft shell it is known as a soft-shell crab (Figure 1Ⅸ). Reviewing the whole moulting process of the mud crab, it was found that moulting caused the crab’s old exoskeleton to be replaced by a larger pliable one, increasing the volume of tissue development inside the body by 30 to 40% [34]. However, the soft-shell crabs will continuously harden the pliable newly originated exoskeleton over time (maybe a few hours or several days). Thus, soft-shell crabs are favoured by consumers in many regions and countries because of their convenience in eliminating the necessity of extracting meat from their hard exoskeleton.

### 3.2. Proximate Composition in Soft-Shell Crab

The effects of sex and moulting stages on the proximate composition (moisture, protein, lipid, and ash) of the edible parts of mud crabs are shown in Figure 2. Two-way ANOVA showed that the moisture and lipid contents in the edible part of the mud crab were significantly affected by the moulting stages (*p* < 0.001), but not by the sex differences (*p* > 0.05). The moisture contents showed a sharply increasing followed by a decreasing trend from the premoult to postmoult Ⅱ stage in male and female crabs, respectively, while the lipid content showed an opposite trend that generally followed the negative correlation phenomenon between lipid and moisture contents [35]. Furthermore, the protein and ash contents in the edible part of the mud crab were significantly affected by moulting stages and sex differences. The protein contents in the edible parts of the female (70.29%, 64.53%, 62.97%, and 65.55%) and male crabs (70.33%, 65.35%, 62.11%, and 68.65%) first showed a decreasing trend and then an increasing trend from the premoult stage to the postmoult Ⅱ stage. However, the ash levels demonstrated an opposite trend where the ash contents in the edible parts of the female and male crabs during ecdysis (24.16% and 25.66%) and postmoult Ⅰ (24.17% and 25.29%) were significantly higher than the premoult (12.17% and 13.36%) and postmoult Ⅱ stages (12.59% and 12.70%) (*p* < 0.001). Previous studies have shown that the proximate composition of the edible portions roughly reflects the nutritional value of crustaceans as food for consumers [3,36], whereas based on the findings of this study, it can be assumed that the soft-shell crab is an ideal food with low lipid (13.99–22.43%) and high protein content (62.11–70.33%).

### 3.3. Mineral Composition in Soft-Shell Crab

The effects of sex and moulting stages on mineral composition (Ca, Mg, Zn, Fe, Cu, Mn, and Se) in the edible parts of mud crabs are shown in Figure 3. It can be noticed that the concentrations of Ca, Mg, Zn, Fe, Cu, Mn, and Se in the edible parts of mud crabs were significantly affected by the moulting stages (*p* < 0.001). The levels of Ca, Mg, Fe, and Cu in the edible parts of female and male crabs significantly increased and then decreased, and the highest values of Ca (1674.87, 1580.22 mg kg^−1^), Mg (834.46, 894.80 mg kg^−1^), Fe (34.33, 22.77 mg kg^−1^), and Cu (17.78, 15.35 mg kg^−1^) were found in the postmoult Ⅰ stage. The concentration of Zn was significantly decreased (from 43.08 to 30.82 mg kg^−1^ in female crabs, from 44.58 to 31.53 mg kg^−1^ in male crabs; *p* < 0.001) with the moulting process; however, the Se level was significantly increased from the premoult stage (1.20 mg kg^−1^ in female crabs, 1.15 mg kg^−1^ in male crabs) to the postmoult Ⅱ stage (1.57 in female crabs, 1.61 mg kg^−1^ in male crabs; *p* < 0.001). Additionally, significantly higher contents of Mn were found in the edible parts of female and male crabs at the postmoult Ⅰ stage (7.71, 7.62 mg kg^−1^) as compared to the other stages. Furthermore, the Ca, Zn, Mn, and Se contents in the edible parts of mud crabs were not significantly influenced by sex (*p* > 0.05). Increased contents of minerals such as Ca, Mg, Fe, Cu, Mn, Zn, and Se might be contributed to the higher ash content in the soft-shell crabs, which indicates that the soft-shell crabs are a mineral-rich aquatic product. Minerals are inorganic elements that are required as essential nutrients for humans to carry out the necessary functions for health and life that are important to prevent malnutrition [37]. The consumption of specific foods that are enriched with mineral contents and required elements are highly recommended and deemed as a proper elemental source for the body [38]. The soft-shell crabs therefore supplement these dietary requirements for human health and are good seafood products with a repertoire of delicacies. Moreover, interestingly in this study, the edible parts of soft-shell crabs in the postmoult Ⅰ stage contained higher levels of mineral elements than those in the ecdysis stage. These findings were consistent with Mohapatra et al., who reported that Ca, Mn, Fe, and Cu were higher and Zn was lower in the soft-shell mud crabs than in the hard-shell crabs [16]. Interestingly, crustaceans generally have a high content of Ca located in the exoskeleton as calcium carbonate [39]. Ca is essential for bone health and other metabolic processes, and calcium-deficient diets are associated with an increased risk of osteoporosis [38], which means that the consumption of soft-shell crabs as a dietary source may be helpful to prevent osteoporosis. Mg is present in the cuticle of crustaceans in high concentrations [40], and as rich sources of haem iron, soft-shell crabs could help to prevent or treat iron deficiency and anaemia [37]. Furthermore, Zn plays a key role in the processes of growth, development, reproduction, cell division, and protein synthesis, which are especially important for infants, children, adolescents, and pregnant women [41]. In this study, the loss of Zn was found in soft-shell crabs, which proved that Zn was utilised during the process of moulting [16]. Therefore, we can sum up that the soft-shell crabs in the postmoult Ⅰ stage were better than those in the other stages, particularly in terms of the nutritional mineral elements.

### 3.4. Total Carotenoid Content in Soft-Shell Crab

The effects of sex and moulting stages on the total carotenoid content in the edible parts of mud crabs are shown in Figure 4. The concentration of total carotenoids was significantly affected by the sex and moulting stages (*p* < 0.001) and increased significantly from the premoult stage to the ecdysis stage, then decreased significantly from the ecdysis stage to the postmoult Ⅱ stage. In female crabs, the total carotenoid content was significantly higher than in male crabs during all moulting stages (*p* < 0.001). Carotenoids include carotenes (simple hydrocarbons mainly in α and β forms) and xanthophylls (derived from carotenes by the addition of oxygen atoms: astaxanthin, canthaxanthin, flavoxanthin, lutein, etc.) [42]. Carotenoids are found in a variety of aquatic animals and can provide a bright red or pink colour to crustaceans [43]. The body colour of commercial crustacean species significantly affects product quality, consumer acceptance, and market prices to a certain extent [44]. In addition, carotenoids can be used as nutritional supplements and antioxidants to prevent diabetes, cardiovascular disease, and neurodegenerative diseases in humans [45]. Hence, considering the total carotenoid content and its physiological efficacy, female soft-shell crabs have an advantage over male soft-shell crabs.

### 3.5. Amino Acid Composition in Soft-Shell Crab

The amino acid composition in the edible parts of male and female crabs during different moulting stages is shown in Figure 5. Most of the amino acids were significantly affected by moulting stages, except Thr. Almost all amino acids in the edible part of the mud crabs were significantly reduced to various degrees from the premoult to postmoult Ⅱ stage. Furthermore, the contents of lysine (Lys), arginine (Arg), histidine (His), threonine (Thr), valine (Val), methionine (Met), phenylalanine (Phe), cysteine (Cys), glycine (Gly), serine (Ser), alanine (Ala), aspartic acid (Asp), glutamic acid (Glu), proline (Pro), and tyrosine (Tyr) were significantly affected by sex differences. For example, in male crabs, the amino acid content of the edible parts was significantly higher than that of female crabs at different moulting stages. The principal component analysis (PCA) score plot (Figure 6A) and loading plot (Figure 6B) of amino acid composition for the edible parts of mud crabs are shown in Figure 6. The first two principal components (PCs) revealed 89.37% variation (Figure 6A; 75.50% and 13.87% of the total variance, respectively). It can be observed that all replicates of the eight treatments were divided into eight intuitively separated clusters (Figure 6A). Amino acids are important molecules with critical roles in protein synthesis, immune and antioxidant responses, energy homeostasis, and health [46]. Furthermore, some functional amino acids (e.g., Glu, Arg, and Lys) can participate in the transport of fatty acids, activation of the oxidation of long-chain fatty acids, and inhibition of fatty acid synthesis [3]. Among these 17 amino acids, Glu had the highest concentration among all moulting stages in the edible parts of female (35.65–46.92 mg g^−1^) and male crabs (63.47–97.68 mg g^−1^). Glu, the major neurotransmitter of the brain, can be involved in cognitive functions such as learning and memory in the brain, and can further increase brain function and mental activity [47]. Another important amino acid, Arg, has many effects on the organism that include the modulation of immune function, wound healing, hormone secretion, insulin sensitivity, and physiological function [48]. Furthermore, Lys is a precursor for the biosynthesis of carnitine as well as a precursor for protein synthesis, which plays an important role in protein synthesis and fatty acid β-oxidation [49]. It was shown that the functional amino acid contents of the edible parts of male crabs was significantly higher than that of female crabs at the corresponding moulting stages. In general, the results of the present study showed that male soft-shell crabs were better than female soft-shell crabs in terms of amino acid nutrition.

### 3.6. Fatty Acid Composition of Soft-Shell Crab

The fatty acid composition in the edible parts of male and female crabs during different moulting stages is shown in Figure 7. Saturated fatty acid (SFA), mono-unsaturated fatty acid (MUFA), n-6 polyunsaturated fatty acid (n-6 PUFA), and n-3 polyunsaturated fatty acid (n-3 PUFA) contents in the edible parts of male and female crabs were significantly affected by moulting stages (*p* < 0.001). Most of the fatty acids were significantly affected by sex differences except 18:2n-6, 20:2n-6, and 20:3n-6, respectively. Interestingly, we found that the EPA (20:5n-3) and DHA (22:6n-3) contents in the edible parts of both female and male crabs in the postmoult II stage were significantly higher than those in the premoult stage. However, the ARA (20:4n-6) content showed a significant gradual decrease from the premoult stage to the postmoult II stage (*p* < 0.001). The PCA analysis provided an illustration of the overall distribution of fatty acid composition in the edible parts of mud crabs. The first two principal components (PCs) revealed 74.40% variation (Figure 8A; 45.63% and 28.77% of the total variance, respectively), representing a significant separation of different groups. Fatty acids, as the most important nutrients, directly affect the nutritional value to a great extent [3]. Many previous studies have suggested that the n-3 long-chain polyunsaturated fatty acids (n-3 LC-PUFA) (particularly EPA and DHA) play important roles in controlling and regulating cell membrane fluidity, lipid metabolism, insulin resistance, and nervous system development [2,50,51]. The decrease in SFA levels in edible parts may be explained by the consumption of SFA as an energy substance during moulting. It has been shown that reducing the intake of SFAs, especially 14:0, may decrease the risk of cardiovascular disease [52]. Furthermore, the consumption of MUFA and PUFA is associated with a lower risk of coronary heart disease [53]. The results showed that the edible parts of both female and male crabs in the postmoult II stage had a significantly higher content of EPA and DHA, but significantly lower contents of SFA than those in other stages. Therefore, from the perspective of human nutrition, products derived from soft-shell crabs (female and male) in the postmoult II stage are healthier for consumers due to the excellent fatty acid composition in the edible portions.

## 4. Conclusions

This study indicated that there were significant differences in the nutritional value of the edible parts between male and female soft-shell crabs at different moulting stages, that is, sex differences and moulting stages affected the nutritional quality (proximate composition, mineral elements, total carotenoid, amino acids, and fatty acids) of soft-shell crabs. The present study proposes that the soft-shell crab is an ideal food with low lipid and high protein content. The female or male soft-shell crabs in the postmoult Ⅰ stage had a much richer mineral element content than those in the other moulting stages, and female soft-shell crabs had an advantage of total carotenoid content over male soft-shell crabs, while male soft-shell crabs had a better performance in amino acid nutrition than female soft-shell crabs. Furthermore, soft-shell crabs in the postmoult Ⅱ stage had a more excellent profile of fatty acids than those in the other stages. Consequently, consumers can selectively determine the best time as well as the sex (female and male) of soft-shell crabs to harvest according to the need for nutritional value. The present study clarified the nutritional value of soft-shell crabs of different sexes and in different moulting stages, revealed the potential commercial value of soft-shell crabs as a new type of seafood, provided a reference and theoretical basis for the diversified cultivation of soft-shell crabs, and further promoted the development of the mud crab industry.

## Figures and Tables

**Figure 1 foods-12-00583-f001:**
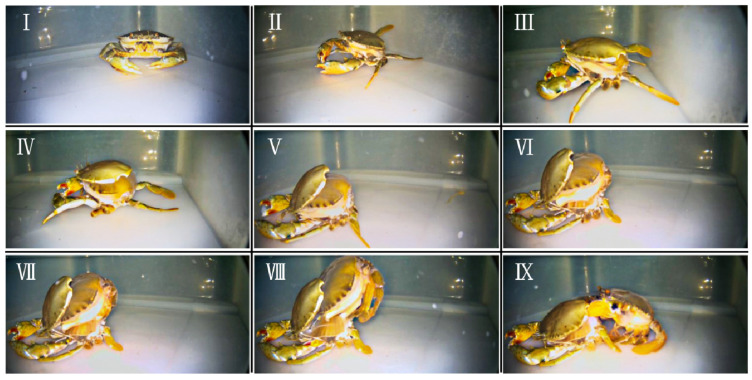
The production process of soft-shell crab (underwater view). (**Ⅰ**–**Ⅱ**) represent the process of premoult. (**Ⅲ**–**Ⅸ**) represent the process of ecdysis.

**Figure 2 foods-12-00583-f002:**
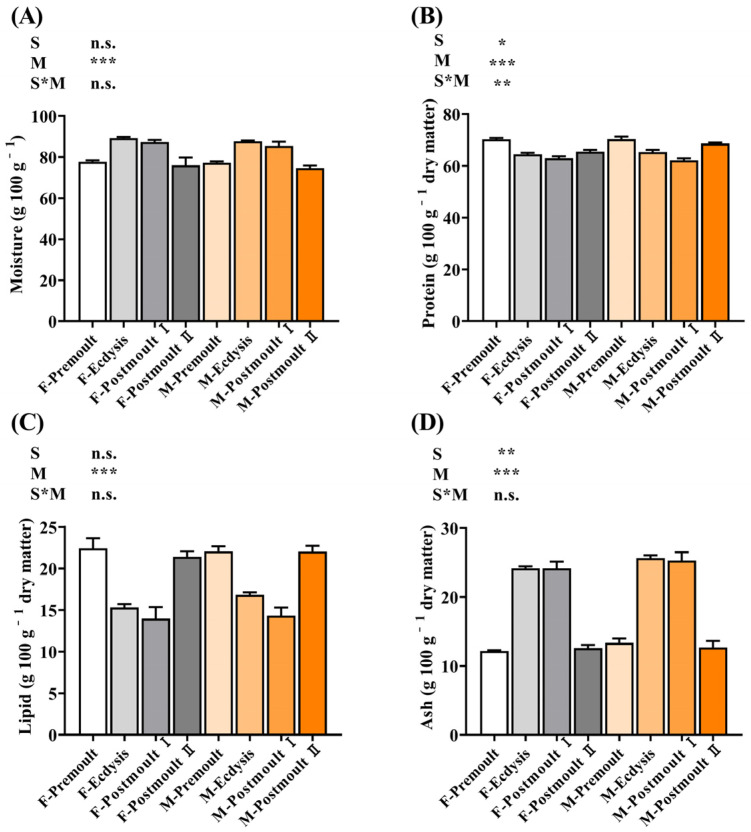
Effects of sex and moult stages on the proximate composition (g 100 g^−1^ dry weight) of the edible part of mud crab. (**A**) moisture; (**B**) protein; (**C**) lipid; (**D**) ash. Columns represent means with bars indicating the standard error (n = 3). Two-way ANOVA *p*-values are shown in each panel, with “S” representing the effects of sex, “M” representing the effects of moulting stages, and “S*M” representing the interaction between sex and moulting stages. n.s., not significant (*p* ≥ 0.05). The differences were considered statistically significant at values of * *p* < 0.05, ** *p* < 0.01, and *** *p* < 0.001.

**Figure 3 foods-12-00583-f003:**
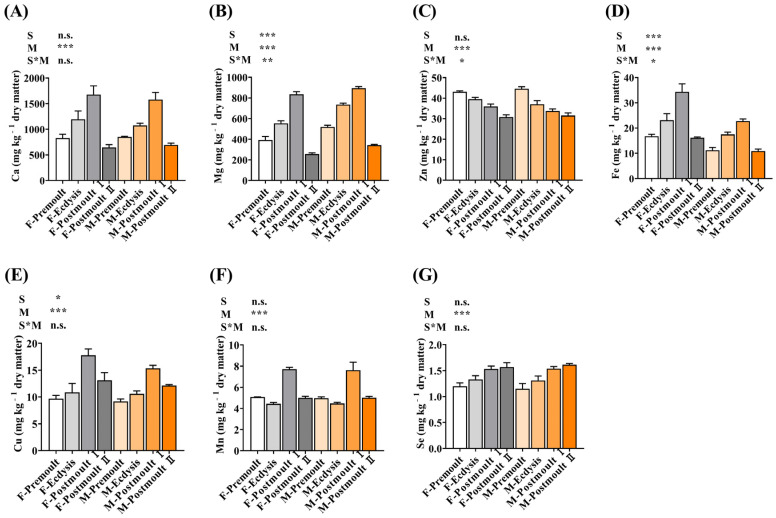
Effects of sex and moult stages on the mineral composition (mg kg^−1^ dry weight) of the edible part of mud crab. (**A**) Ca; (**B**) Mg; (**C**) Zn; (**D**) Fe; (**E**) Cu; (**F**) Mn; (**G**) Se. Columns represent means with bars indicating the standard error (n = 3). Two-way ANOVA *p*-values are shown in each panel, with “S” representing the effects of sex, “M” representing the effects of moulting stages, and “S*M” representing the interaction between sex and moulting stages. n.s., not significant (*p* ≥ 0.05). The differences were considered statistically significant at values of * *p* < 0.05, ** *p* < 0.01, and *** *p* < 0.001.

**Figure 4 foods-12-00583-f004:**
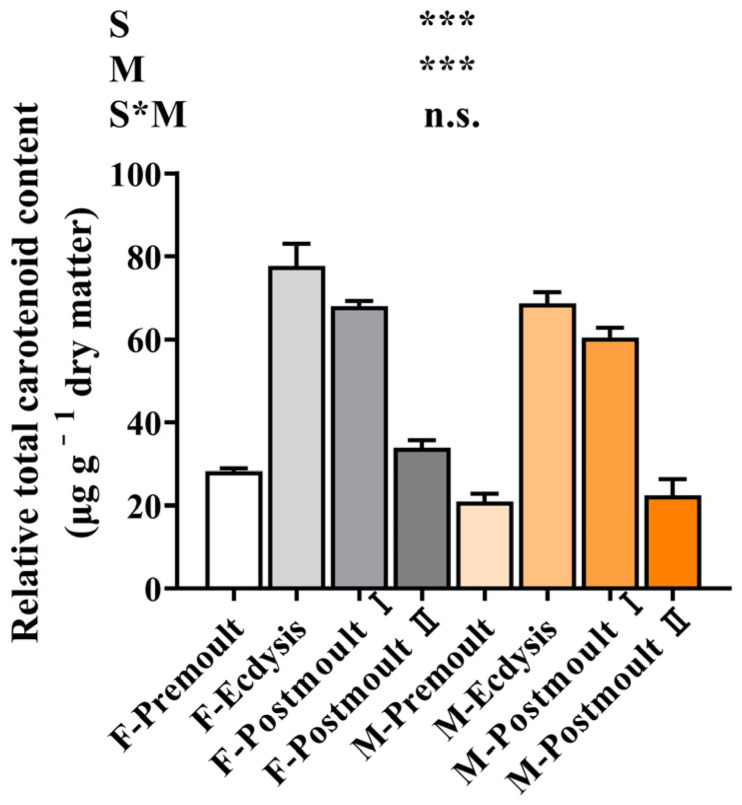
Effects of sex and moult stages on the total carotenoid content (μg g^−1^ dry weight) of the edible part of mud crab. Columns represent means with bars indicating a standard error (n = 3). Two-way ANOVA *p*-values are shown in each panel, with “S” representing the effects of sex, “M” representing the effects of moulting stages, and “S*M” representing the interaction between sex and moulting stages. n.s., not significant (*p* ≥ 0.05). The differences were considered statistically significant at values of *** *p* < 0.001.

**Figure 5 foods-12-00583-f005:**
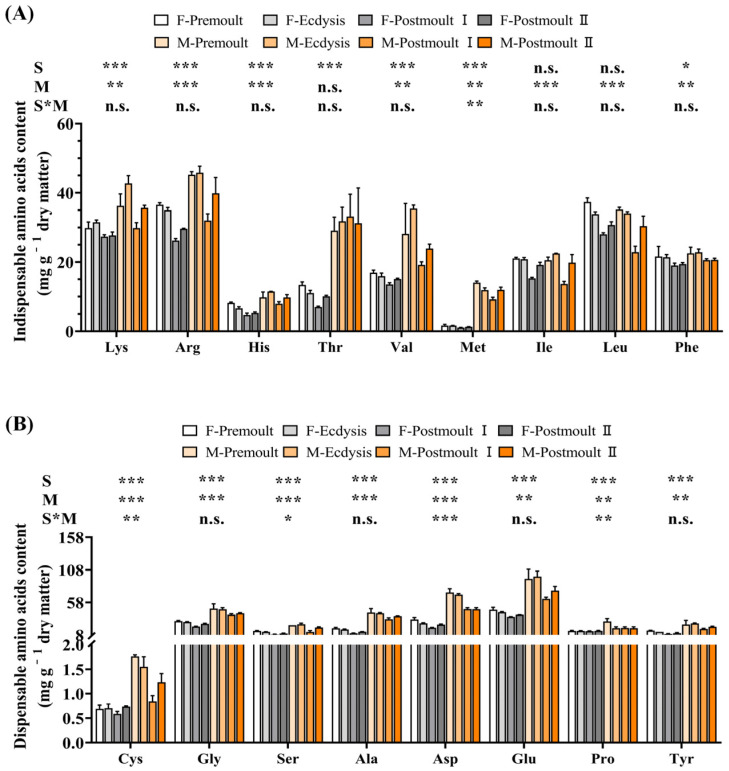
Effects of sex and moult stages on the amino acid composition (mg g^−1^ dry weight) of the edible part of mud crab. (**A**) indispensable amino acids; (**B**) dispensable amino acids. Columns represent means with bars indicating the standard error (n = 3). Two-way ANOVA *p*-values are shown in each panel, with “S” representing the effects of sex, “M” representing the effects of moulting stages, and “S*M” representing the interaction between sex and moulting stages. n.s., not significant (*p* ≥ 0.05). The differences were considered statistically significant at values of * *p* < 0.05, ** *p* < 0.01, and *** *p* < 0.001.

**Figure 6 foods-12-00583-f006:**
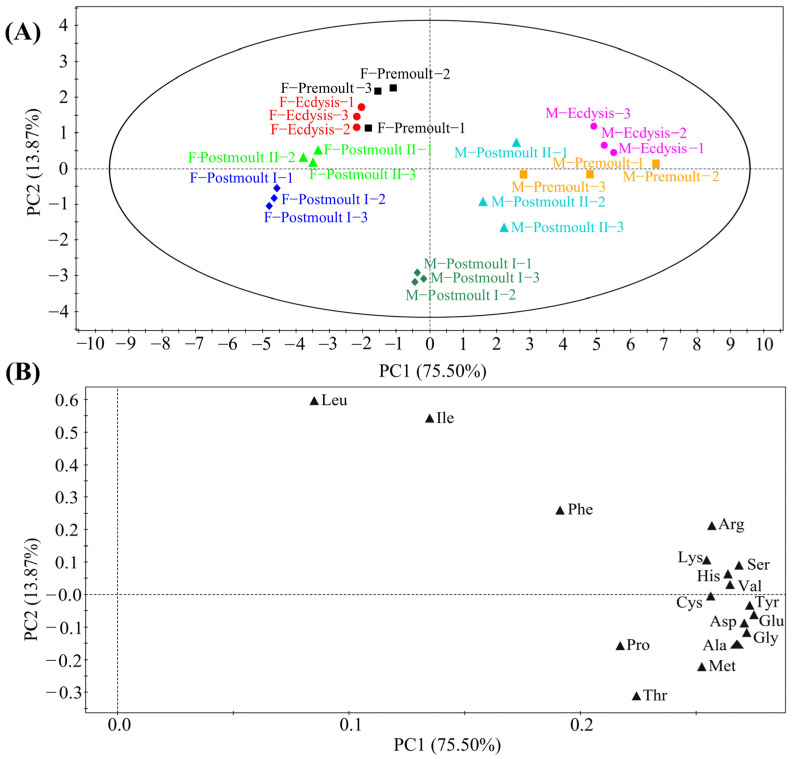
Principal component analysis (PCA) score plot (**A**) and loading plot (**B**) based on the amino acid composition of mud crab.

**Figure 7 foods-12-00583-f007:**
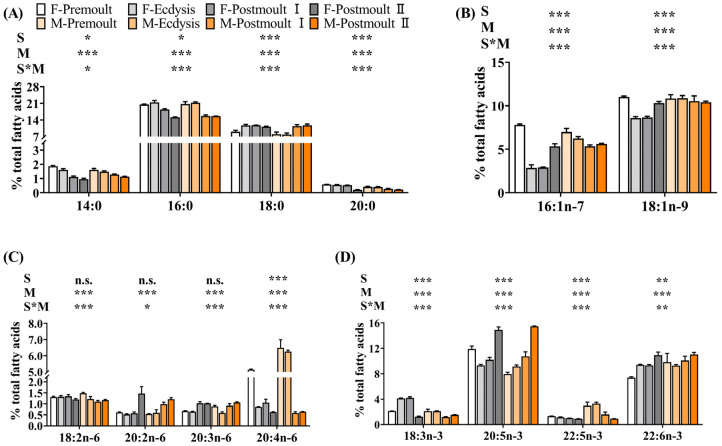
Effects of sex and moult stages on the fatty acid composition (% total fatty acids) of the edible part of mud crab. (**A**) SFA, saturated fatty acids, such as 14:0, 16:0, 18:0, and 20:0; (**B**) MUFA, mono-unsaturated fatty acids, such as 16:1n-7 and 18:1n-9; (**C**) n-6 PUFA, n-6 polyunsaturated fatty acids, such as 18:2n-6, 20:2n-6, 20:3n-6 and 20:4n-6; (**D**) n-3 PUFA, n-3 polyunsaturated fatty acids, such as 18:3n-3, 20:5n-3, 22:5n-3 and 22:6n-3. Columns represent means with bars indicating the standard error (n = 3). Two-way ANOVA *p*-values are shown in each panel, with “S” representing the effects of sex, “M” representing the effects of moulting stages, and “S*M” representing the interaction between sex and moulting stages. n.s., not significant (*p* ≥ 0.05). The differences were considered statistically significant at values of * *p* < 0.05, ** *p* < 0.01, and *** *p* < 0.001.

**Figure 8 foods-12-00583-f008:**
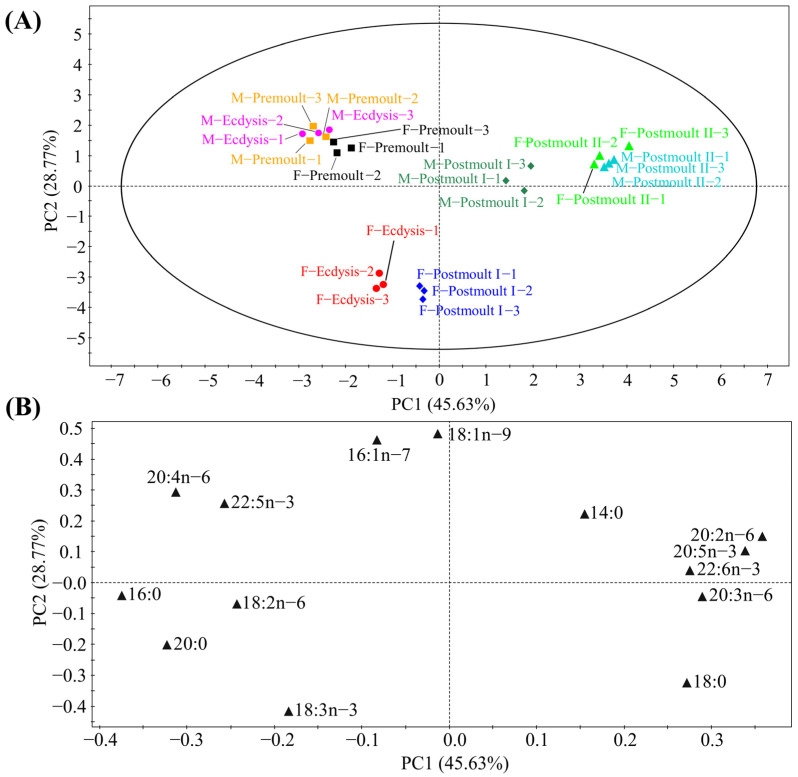
Principal component analysis (PCA) score plot (**A**) and loading plot (**B**) based on the fatty acid composition of mud crab.

## Data Availability

Data are available from the corresponding author upon request.

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
