# Peer review of "Evaluation of the Feasibility of Harvest Optimisation of Soft-Shell Mud Crab (Scylla paramamosain) from the Perspective of Nutritional Values"

_foods, 2023, doi:10.3390/foods12030583_

Round 1

Reviewer 1 Report

Evaluation of feasibility of diversified production of soft-shelled mud crab (Scylla paramamosain) from the perspective of nutritional values

This manuscript investigated the difference in the nutritional vale of the moult stages of the mud crab with respect to harvest optimisation. The data presented was very comprehensive and covers the components suitable for the investigation, however further studies would be required to test if these results translate to commercial conditions. The results presented were reliable and look good for an initial evaluation.

Title

1.       Title of the manuscript could be misleading as to the content of the manuscript. The content would be better represented by a title focusing more towards harvest optimisation based on nutritional value.

Abstract

2.       Fatty Acid Abbreviations in the abstract should have full name and follow with its abbreviation in parentheses.

Section 2.9 Determination of fatty acid composition

3.       At line 197, there seems to be a section of the method misrepresented. It is more likely that the supernatant removed in line 196 was reduced to dryness rather than the pellet remaining in the original tube.

4.       At Line 210 they talk about comparing the Retention Time data with Mass Spectra for identification of the individual fatty acids. This suggests they are comparing GC-FID data to GC-MS Data.

Figures

5.       Figures 3,6,7 and 9 are difficult to read. Adjust the size and fonts of the figures to match Figure 2.

Author Response

Comments and Suggestions for Authors

Evaluation of feasibility of diversified production of soft-shelled mud crab (Scylla paramamosain) from the perspective of nutritional values

This manuscript investigated the difference in the nutritional vale of the moult stages of the mud crab with respect to harvest optimisation. The data presented was very comprehensive and covers the components suitable for the investigation, however further studies would be required to test if these results translate to commercial conditions. The results presented were reliable and look good for an initial evaluation.

Title

  1. Title of the manuscript could be misleading as to the content of the manuscript. The content would be better represented by a title focusing more towards harvest optimisation based on nutritional value.

Response: Thanks for the reviewer’s suggestion, and we modified the title to make it more towards harvest optimisation based on nutritional value. Now the title is “Evaluation of the feasibility of harvest optimisation of soft-shelled mud crab (Scylla paramamosain) from the perspective of nutritional values”.

Abstract

  1. Fatty Acid Abbreviations in the abstract should have full name and follow with its abbreviation in parentheses.

Response: Thanks for the reviewer’s suggestion, and we have added the full name of fatty acids, eicosapentaenoic acid (EPA), docosahexaenoic acid (DHA) and saturated fatty acids (SFA).

Section 2.9 Determination of fatty acid composition

  1. At line 197, there seems to be a section of the method misrepresented. It is more likely that the supernatant removed in line 196 was reduced to dryness rather than the pellet remaining in the original tube.

Response: Thanks for the reviewer’s suggestion, and we have revised these sentences to avoid being misunderstood. “Next, the solution was centrifuged at 3000 rpm at 4 ℃ for 5 min and the supernatant was transferred to a new centrifuge tube. The solution in the tube was reduced to dryness using a termovap sample concentrator.” (Line 205-207)

  1. At Line 210 they talk about comparing the Retention Time data with Mass Spectra for identification of the individual fatty acids. This suggests they are comparing GC-FID data to GC-MS Data.

Response: Thanks for the reviewer’s suggestion, and we have revised the description of the method in the fatty acid identification section.

Figures

  1. Figures 3,6,7 and 9 are difficult to read. Adjust the size and fonts of the figures to match Figure 2.

Response: Thanks for the reviewer’s suggestion, and we have adjusted the size and fonts of the figures 3,6,7,9 to match Figure 2.

Reviewer 2 Report

Manuscript titled “Evaluation of feasibility of diversified production of soft- 2

shelled mud crab (Scylla paramamosain) from the perspective of 3 nutritional values” assessed and compared nutritional factors of mud crab at different molting stages. Furthermore, soft-shell crab production process footage was recorded and visually presented. The MS is well-written, well conceptualized and MM and Results are well presented. I’ve couple of suggestions for the Authors. I kindly invite them to take those into consideration.

At each molding stage tissue samples from 3 crabs were commingled and analyzed as a single sample. How did you make the statistical comparisons without replicates. Did you perform the analysis separately for a single sample multiple time? Please clarify.

Figures are visually satisfying but it would be great If you consider providing values as a supplementary test which I believe it would be helpful for those who might want to compare their nutritional results with your results.

I also would like you to consider discussing and comparing the proximate protein content and sum of amino acid content. For crustaceans Nx6.25 might not be given exact protein content of crab since there are different N sources beside amino acids. You may come up with another coefficient than 6.25 and could represent it for those who’d like to assess crude protein in that crab species.

Please also consider given general values (ranges, important values etc. ) in text as well. It is time consuming to assess the range of certain things like crude protein content, mineral content etc.

Author Response

Comments and Suggestions for Authors

Manuscript titled “Evaluation of feasibility of diversified production of soft-shelled mud crab (Scylla paramamosain) from the perspective of 3 nutritional values” assessed and compared nutritional factors of mud crab at different molting stages. Furthermore, soft-shell crab production process footage was recorded and visually presented. The MS is well-written, well conceptualized and MM and Results are well presented. I’ve couple of suggestions for the Authors. I kindly invite them to take those into consideration.

At each molding stage tissue samples from 3 crabs were commingled and analyzed as a single sample. How did you make the statistical comparisons without replicates. Did you perform the analysis separately for a single sample multiple time? Please clarify.

Response: Thanks for the reviewer’s suggestion. In fact, whatever female and male crabs, each molt stage (premolt, ecdysis, postmolt â…  and postmolt â…¡) contained 10 crabs, respectively. Each treatment group had three replicates, with each replicate containing three crabs randomly (Line 125-133, Line 135-143). In addition, we performed the analysis separately for a single sample multiple time. All determinations performed in triplicate and the coefficient of variation was within 1.0%. Each value from replicate was the average of the three measurements. We have made relevant statements in each test items in the text.

Figures are visually satisfying but it would be great If you consider providing values as a supplementary test which I believe it would be helpful for those who might want to compare their nutritional results with your results.

Response: Thanks for the reviewer’s suggestion. The amount of data involved in this study is very large, and the images present a good comparison effect at a glance, which also allows readers to have a very intuitive concept of the nutritional value of male and female green crabs at different molting stages. According to the reviewer's opinion, we have provided some key values (such as protein, amino acid and fatty acid, etc.), so that readers can have a better understanding of the range of important nutritional indicators.

I also would like you to consider discussing and comparing the proximate protein content and sum of amino acid content. For crustaceans Nx6.25 might not be given exact protein content of crab since there are different N sources beside amino acids. You may come up with another coefficient than 6.25 and could represent it for those who’d like to assess crude protein in that crab species.

Response: Thanks for the reviewer’s suggestion and question. This is a particularly neglected problem, especially in the testing of nutritional quality of food. The total amino acid contents of the edible part in mud crab ranged from approximately 260.90 mg/g (26.09%) to 585.24 mg/g (58.52%) (dry matter) among the treatments based on the original data. While the crude protein content ranged from 62.11 % to 70.33 % in the edible part of crab, which were higher than the value of total amino acid contents. This is understandable because in addition to amino acid nitrogen, crude proteins contain other sources of nitrogen when measured. However, other methods are needed to determine the amino nitrogen and non-amino nitrogen, then we may probably estimate the approximate coefficient. Here, we still don't have an exact number for readers to refer to.

Please also consider given general values (ranges, important values etc. ) in text as well. It is time consuming to assess the range of certain things like crude protein content, mineral content etc.

Response: Thanks for the reviewer’s suggestion. We have provided the general values of certain things like crude protein content, mineral content etc in the text.